# MR Image Restoration by Utilizing Fractional-Order TV and Recursive Filtering

Nana Wei, Wei Xue[(⊠)]

*School of Computer Science and Technology*
*Anhui University of Technology*
Maanshan 243032, China
Email: nn.wei@foxmail.com, xuewei@ahut.edu.cn

*Abstract*—**Total variation based methods are effective models in magnetic resonance image restoration. For eliminating impulse noise, an effective way is to use the $\ell_0$-norm total variation model. However, the TV image restoration consistently produces staircase artifacts, particularly at noise levels with high density. In this paper, we propose a novel MR image restoration model incorporating fractional-order regularization and filtering methods. Specifically, the first term employs the $\ell_0$-norm as the data fidelity term to effectively eliminate impulse noise. The second term introduces a fractional-order total variation regularizer, which preserves structural information while mitigating staircase artifacts during deblurring. Given its suboptimal performance in texture detail recovery, we utilize recursive filtering for high-quality edge-preserving filtering. Finally, we solve the corresponding optimization model by using the alternating direction method of multipliers. Experimental results demonstrate the effectiveness of our method in restoring medical images.**

*Index Terms*—**Magnetic resonance image restoration, $\ell_0$-norm, fractional-order total variation, recursive filtering**

## I. INTRODUCTION

Magnetic Resonance (MR) Image is a powerful tool used in the field of medical imaging for the diagnosis and monitoring of various diseases. However, the quality of MR images can often be compromised by the presence of impulse noise, also known as impulse noise. This type of noise appears as sparsely occurring white and black pixels, and can significantly degrade the image quality, making it difficult for medical professionals to interpret the images accurately. Impulse noise in MR images can be caused by a variety of factors, including hardware faults, transmission errors, and other interferences. The challenge lies in developing effective methods to remove this noise while preserving the important details and edges in the images.

To remove the impulse noise, a spectrum of approaches has been explored. Among them, regularization methods and filtering methods are widely recognized. A cornerstone in the realm of medical image restoration is the Total vriation (TV) model, which has demonstrated remarkable efficacy in denoising, deblurring, segmentation, and superresolution tasks. However, the TV model encounters limitations when confronted with noise in piecewise affine regions, often manifesting as undesirable staircase artifacts. To overcome this issue, a series of regularization functions has been proposed. Such as, the High-Order TV method [3], Non-Convex High-Order TV [4], Fractional-Order TV (FOTV). [5] and [6] are among the notable advancements, their numerical experiments showed that their methods alleviated the staircase effect effectively. Besides, [2] proposed the Overlapping Group Sparse Fractional-Order TV (OGSTV), which has garnered attention for its refined approach. Besides, [10] proposed an enhanced $\ell_1$-NCFOTV method, which promises to elevate the restoration precision of images afflicted by impulse noise and to diminish the staircase effect.

In the filtering domain, the Median Filter (MF) and Adaptive Median Filter (AMF) have been mainstays since the early days of denoising endeavors. Despite their popularity, these filters falter under the weight of high noise intensity, prompting the evolution of more sophisticated filtering techniques such as Recursive Filtering (RF), Recursive Mean Filtering, A non-local adaptive mean filter(NAMF) [8] and improved median filter [9]. Additionally, recursive filtering demonstrates effective performance in restoring edges.

To effectively remove impulse noise, the $\ell_2$-norm [1] is sensitive to outliers and can easily result in unsatisfactory image restoration, which is commonly used to restore images degraded by additive Gaussian noise. Furthermore, there has been a growing interest in using an $\ell_1$-fidelity term instead of the $\ell_2$-fidelity term for image restoration in many literature such as [11], [12]. The numerical evidence clearly indicates that the proposed method has made substantial advancements in its capacity for restoration. While the $\ell_1$-norm has been recognized for its considerable benefits in the realms of sparse signal processing and image restoration, it has been noted that it might excessively penalize the resulting solution in the context of impulse noise elimination, as highlighted by [7].

To circumvent this issue, a method for removing impulse noise using $\ell_0$ total variation ($\ell_0$-TV) was proposed in [13]. This method employs the $\ell_0$-norm for the data fidelity and uses the proximal alternating direction method of multipliers for image restoration. The empirical evidence show better performance than $\ell_1$-norm based methods. Consequently, this indicates that $\ell_0$-norm is suitable for the restoration of images distorted by impulse noise. The image restoration of the

This work was supported in part by the Natural Science Foundation of the Anhui Higher Education Institutions of China (Grant Nos. 2023AH040149 and 2022AH050310) and the Anhui Provincial Natural Science Foundation (Grant No. 2208085MF168).

optimization problem can be depicted as:

$$\min_{u} \|o \odot (Hu - b)\|_0 + \lambda\phi_{TV}(u), \quad (1)$$

where $u \in \mathbb{R}^{n \times m}$ is the desired original clean image, $b \in \mathbb{R}^{n \times m}$ is the degraded image, $\lambda > 0$ is the regularization parameter, $o \in \{0,1\}^n$ is specified by the user, $\odot$ denotes an elementwise product, $H \in \mathbb{R}^{n \times n}$ is a linear operator. Another circumstance, when $H = I$, the identity operator, it translates into a denoising issue.

In recent work, the $\ell_0$-norm has been employed as a data fidelity criterion for the elimination of impulse noise. [2] proposed the $\ell_0$-norm with an OGSTV and MC penalty model, which can significantly enhance the model's proficiency in image restoration, particularly under conditions characterized by high levels of impulse noise. [14] proposed $\ell_0$-OGSTV model effectively diminishes staircasing artifacts and precisely restores images marred by impulse noise. [15] has successfully integrated the $\ell_0$-norm data fidelity term with a nonconvex generalized regularization approach. This integration enhances the model's ability to preserve sharp image contours while simultaneously reducing staircase artifacts. Collectively, these models showcase the effectiveness of the $\ell_0$-norm in restoring images that have been corrupted by impulse noise.

To the best of our knowledge, the combination of $\ell_0$-norm data fidelity, fractional-order total variation and recursive filtering has not been addressed by any of the existing optimization models. Therefore, this has motivated us to combine them for the restoration of MR images in medical imaging. In this article, we focus on addressing the challenge of restoring MR image in medical imaging that have been degraded by blurred boundaries and residual noise. The model comprises an $\ell_0$-norm data fidelity term to restore images from impulse noise, a regularizer of Fractional-Order total variation (FOTV) to eliminate the staircase effects and the recursive filtering (RF) to improve edge quality significantly. Our method capitalizes on the strengths of the Forward-Backward Total Variation to provide superior denoising capabilities, effectively mitigating the staircasing artifacts. Additionally, the recursive filtering technique employed not only refines the texture details within medical images, but also reduces the computational process. we employ the alternate direction multiplier algorithm to solve the subproblems. Finally, we conduct numerical experiments to analyze the effectiveness of our proposed model.

The rest of this article is organized as follows. In Section 2 introduces fundamental ideas and preliminary information pertinent to the proposed algorithm. In Section 3, We propose a novel framework aimed at eliminating blur and impulse noise and derive an efficient algorithm to solve the corresponding minimization problem. In Section 4, the superiority of the proposed method is proved by numerical experiments. Finally, a conclusion is made in Section 5.

## II. Preliminaries

In this section, we briefly review a few key components of the $\ell_0$-norm fidelity term, the discrete fractional-order difference, the recursive filtering and the Alternating Direction Method of Multipliers framework.

### A. The $\ell_0$ Fidelity Term

Initially, we provide an overview of the essential definitions and characteristics associated with $\ell_0$-norm fidelity term. In the expression $\|o \odot (Hu - f)\|_0$, the $o \in \{0,1\}^n$ is specified by the user. Specifically, when $o_i = 0$, it implies the pixel in position $i$ is an outlier, while when $o_i = 1$, it implies the pixel in position $i$ is a potential outlier. For this paper, we set $o_i = \begin{cases} 0, & t_i = u_{min} \ or \ u_{max} \\ 1, & otherwise \end{cases}$ for the salt-and-pepper impulse noise.

The following lemma, as meticulously outlined in the work of [13], elucidates the variational formulation that underpins the $\ell_0$-norm.

*Lemma 1:* For any given $w \in \mathbb{R}^n$, it holds that

$$\|w\|_0 = \min_{0 \le z \le 1} <1, 1 - z>, \quad s.t. \ z \odot |w| = 0, \quad (2)$$

and $z^* = 1 - sign(|w|)$ is the unique optimal solution to problem (2). Here, the standard signum function sign is employed in component form, and $sign(0) = 0$.

The result of Lemma 1 implies that the $\ell_0$-norm minimization problem in (1) is equivalent to

$$\min_{0 \le u, z \le 1} <1, 1 - z> + \lambda \|\nabla u\|_1$$
$$s.t. \ z \odot |o \odot (Hu - b)| = 0, \quad (3)$$

if $u^*$ is an overall optimal solution of (1), then $(u^*, 1 - sign(|Hu^* - b|))$ is overall optimal solution of (3). In a similar manner, if $u^*$ is a global Ideal solution of (3), then $(u^*, 1 - sign(|Hu^* - b|))$ is an overall optimal solution of (1). The ability of $\ell_0$-norm has proven to be remarkably effective in restoring images, offering superior performance.

### B. Fractional-Order Total Variation

Given an image domain $\Omega \subset \mathbb{R}^2$, we discretize it as a rectangular grid $\{(x_i, y_j) : 1 \le i \le M, 1 \le j \le N\}$, Consequently, the image discretized on the Euclidean plane $\mathbb{R}^{M \times N}$, denoted as $u_{i,j} = u(x_i, y_j)$. Let $C_0^\alpha(\Omega)$ with $\alpha > 0$ be the space of $\alpha$-order continuously differentiable functions defined on $\Omega$ with compact support. Based on the GL fractional-order derivatives, Thus, the discrete form of the fractional-order gradient $\nabla^a u$ can be evaluated by

$$\nabla^\alpha u = [D_x^\alpha u, D_y^\alpha u], \quad (4)$$

where $\alpha$ is the fractional order and we set $1 \le \alpha < 2$ in this paper. where the fractional-order derivatives $(D_x^\alpha u)$, $(D_y^\alpha u) \in \mathbb{R}^{M \times N}$ along the $x$-axis and the $y$-axis are approximated by the

$$\begin{cases} (D_x^\alpha u)_{i,j} = \sum_{k=0}^{K-1} (-1)^k C_k^\alpha u_{i-k,j} \\ (D_y^\alpha u)_{i,j} = \sum_{k=0}^{K-1} (-1)^k C_k^\alpha u_{i,j-k} \end{cases} \quad (5)$$

In this context, $K$ denotes the quantity of adjacent pixels employed to calculate the fractional-order derivative for

each pixel. The coefficients $\{C_k^\alpha\}_{k=0}^{K-1}$ are given by $C_k^\alpha = \frac{\Gamma(\alpha+1)}{\Gamma(k+1)\Gamma(\alpha+1-k)}$ with the Gamma function $\Gamma(x)$. Hence, the discrete fractional order TV of $u$ is expressed in terms of

$$\|\nabla^\alpha\|_1 = \sum_{i,j}(|(D_x^\alpha u)_{i,j}| + |(D_y^\alpha u)_{i,j}|), \qquad (6)$$

where $(\nabla^\alpha)^* = \overline{(-1)^\alpha}div^\alpha$ is the conjugate operator of the fractional order gradient operator. In the discrete case, the vector $div^\alpha p = (p^{(1)}, p^{(2)}) \in \mathbb{R}^{N \times M} \times \mathbb{R}^{N \times M}$ discrete fractional-order divergence is defined as [6], [17]

$$(div^\alpha p)_{i,j} = (-1)^\alpha \sum_{k=0}^{K-1}(-1)^k C_k^\alpha (p_{i+k,j}^{(1)} + p_{i,j+k}^{(2)}). \quad (7)$$

Observe that the divergence (7) is the adjoint of the gradient (4).

### C. Recursive Filtering

The first-order recursive filtering (RF) was initially introduced by [16]. Denoting $I[u]$ and $J[u]$ are the noisy image and the denoising image, respectively. RF computes $J[u]$ recursively.

$$J[u] = (1-a)^{d[u]}I[u] + a^{d[u]}J[u-1], \qquad (8)$$

where $a \in [0,1]$ is a feedback coefficient, and its implementation in $O(N)$ time is straight forward. Besides the scalar constant $a = exp(-\sqrt{2}/\sigma_s)$ is a user defined parameter controlling the relative emphasis of $I[u]$ and $J[u]$.

$$d[u] = 1 + \frac{\sigma_s}{\sigma_r}|I[u] - I[u-1]| = 1 + \frac{\sigma_s}{\sigma_r}|\nabla I[u]|, \quad (9)$$

where $\sigma_s$ and $\sigma_r$ represent the spatial and range parameters, respectively. The rapid iteration of the recursion in (8) is attributed to the pre-computed values from (9) and the independent computation of $J[u]$ for each row. For gray images, one can sequentially perform (8) forward and backward in two directions. With color images, it's necessary to iterate through this operational sequence for each color channel.

As $d$ increases, $a^d$ goes to zero, stopping the propagation chain, thus preserving edges. Furthermore, the expanded recursion of Eq. (8) also elucidates this issue

$$J[u] = \sum_{\ell=0}^{n}\left(\prod_{k=0}^{\ell}a^{d[u-k+1]}\right)(1 - a^{d[u-\ell]})I[u-\ell]. \quad (10)$$

### D. Alternating Direction Method of Mltipliers

The alternative direction method of multipliers (ADMM) is a computational framework for solving optimization problems , which is to solve the following constrained separable optimization problems:

$$\min_{u,w} f(u) + g(w) \quad s.t. \ Au + Bw = d, \ u, w \in \chi_i, i = 1, 2, \tag{11}$$

where $f(\cdot)$, $g(\cdot) : \chi_i \to \mathbb{R}$ are closed convex functions, $A$, $B \in \mathbb{R}^{l \times n}$ are linear transforms, $\chi_i \to \mathbb{R}$ are nonempty closed convex sets, and $d \in \mathbb{R}^l$ is a given vector. For problem (11), we establish the augmented Lagrangian function

$$\begin{aligned}\mathcal{L}_\mathcal{A}(u, w; \mu) =& f(u) + g(w) + \mu^T(Au + Bw - d) \\ &+ \frac{\lambda}{2}\|Au + Bw - d\|_2^2.\end{aligned} \tag{12}$$

where $\mu \in \mathbb{R}^l$ is the Lagrange multiplier and $\lambda > 0$ is a penalty parameter which controls the linear constraint. The objective is to find the saddle point of $\mathcal{L}_\mathcal{A}$ by alternatively minimizing $\mathcal{L}_\mathcal{A}$ with respect to $u$, $w$ and $\mu$.

The problem (11) is addressed by presenting the ADMM algorithm as Algorithm 1.

---

**Algorithm 1** ADMM for minimizing the problem (11).

**Input:** penalty parameter $\lambda > 0$, number of iterations.
**Initialize:** Initial image $u^0 = b$, counter $k = 0$, Lagrange multipliers $\mu$.
**Output:** Restored image $u$.
1: For $k = 0$, compute $u^{k+1}$, $w^{k+1}$, $\mu^{k+1}$
2: $u^{k+1} = \arg\min_u f(u) + \frac{\lambda}{2}\|Au + Bw^k - d + \frac{\mu^k}{\lambda}\|_2^2$,
3: $w^{k+1} = \arg\min_w g(w) + \frac{\lambda}{2}\|Au^{k+1} + Bw - d + \frac{\mu^k}{\lambda}\|_2^2$,
4: $\mu^{k+1} = \mu^k + \lambda(Au^{k+1} + Bw^{k+1} - d)$,
5: $k = k + 1$
6: until a stopping criterion is satisfied.

---

## III. THE PROPOSED ALGORITHM

In this part, we first introduce the proposed MR image restoration model and the corresponding solution methods. Finally, an ADMM solution framework is provided.

### A. Model

The proposed MR image restoration model is as follows

$$\min_{0 \leq u \leq 1}\|o \odot (Hu - b)\|_0 + \lambda_1\|\nabla^\alpha u\| + \lambda_2\phi(u), \tag{13}$$

where $\nabla^\alpha u$ denotes the fractional-order TV. $\phi$ denotes the recursive filtering. $\lambda_1 > 0$ and $\lambda_2 > 0$ represent regularization parameters.

### B. Optimization

Using variable splitting, the problem is rephrased as a constrained optimization problem that follows

$$\begin{aligned}\min_{0 < u, v \leq 1} \ &< 1, 1 - v > + \lambda_1\|x\| + \lambda_2(z) \\ s.t. &Hu - b = y \\ &v \odot |o \odot y| = 0 \\ &\nabla^\alpha u = x, u = z.\end{aligned} \tag{14}$$

The corresponding augmented Lagrangian functional is given by

$$
\begin{aligned}
\mathcal{L}_\mathcal{A}&(u,v,x,y,z,\mu_v,\mu_x,\mu_y,\mu_z)\\
&=<1,1-v>+\lambda_1||x||+\lambda_2\phi(z)\\
&\quad+<v\odot o\odot|y|,\mu_v>+\frac{\beta_v}{2}||v\odot o\odot|y|||^2\\
&\quad+<Hu-b-y,\mu_y>+\frac{\beta_y}{2}||Hu-b-y||^2\\
&\quad+<\nabla^\alpha u-x,\mu_x>+\frac{\beta_x}{2}||\nabla^\alpha u-x||^2\\
&\quad+<u-z,\mu_z>+\frac{\beta_z}{2}||u-z||^2,
\end{aligned}
\tag{15}
$$

where variables $\mu_v$, $\mu_x$, $\mu_y$, and $\mu_z$ are the Lagrange multipliers associated with the constraints of (14). $\beta_v, \beta_x, \beta_y$, and $\beta_z > 0$ are the corresponding penalty parameters. We utilize the alternating direction method of multipliers [18] to solve the proposed model (15). According to the ADMM scheme, we can alternately solve for the following problems.

*1) u-subproblem:* The u-subproblem

$$
\begin{aligned}
u^{k+1}=\arg\min_u&\frac{\beta_y}{2}||Hu-b-y+\frac{\mu_y}{\beta_y}||^2\\
&+\frac{\beta_x}{2}||\nabla^\alpha u-x+\frac{\mu_x}{\beta_x}||^2\\
&+\frac{\beta_z}{2}||u-z+\frac{\mu_z}{\beta_z}||^2.
\end{aligned}
\tag{16}
$$

Based on the first-order optimality conditions, we are tasked with resolving a system of linear equation

$$
\begin{aligned}
u(\beta_y H^T H+\beta_x(\nabla^\alpha)^T\nabla^\alpha)+\beta_z)=\\
H^T(\beta_y(b+y)-\mu_y)+(\nabla^\alpha)^T(\beta_x x-\mu_x)+\beta_z(z-\mu),
\end{aligned}
\tag{17}
$$

where $\mu=\frac{\mu_z}{\beta_z}$, considering $u$ with periodic boundary constraints. Due to the circulant and circulant blocks (BCCB) structure, matrices $(\nabla^\alpha)^T\nabla^\alpha$ and $H^T H$ can be diagonalized by 2D discrete fast Fourier transforms ($FFT$). Therefore, solving for $u$ can be efficiently solved using 2D $FFT$ and 2D $FFT$ inverse operations. The process for acquiring the optimal $u$ is outlined as follows

$$
u^{k+1}=\mathcal{F}^{-1}\left(\frac{\mathcal{F}(\varkappa)}{\mathcal{F}[\beta_y H^T H+\beta_x(\nabla^\alpha)^T(\nabla^\alpha)+\beta_z]}\right),
\tag{18}
$$

where $\varkappa=H^T(\beta_y(b+y)-\mu_y)+(\nabla^\alpha)^T(\beta_x x+\mu_x)+\beta_z(z-\mu)$, $\mathcal{F}$ and $\mathcal{F}^{-1}$ represent the Fourier transform and its inverse.

*2) v-subproblem:* The $v$ subproblem can be written as

$$
\begin{aligned}
v^{k+1}=\arg\max_v&\langle1,1-v\rangle+<v\odot o\odot|y^k|,{\mu_v}^k>\\
&+\frac{{\beta_v}^k}{2}||v\odot o\odot|y^k|||^2.
\end{aligned}
\tag{19}
$$

The $v$-subproblem in (19) is equivalent to

$$
\begin{aligned}
v^{k+1}=\arg\max_v&\frac{1}{2}\beta_v o\odot y^k\odot y^k\odot v^2\\
&+v\left(\mu_v^k\odot o\odot\left|y^k\right|-1\right),
\end{aligned}
\tag{20}
$$

therefore, projection method is engaged to find the solution

$v^{k+1}$

$$
v^{k+1}=\min\left(1,\max\left(0,-\frac{\mu_v^k\odot o\odot|y^k|-1}{\beta_v o\odot y^k\odot y^k}\right)\right).
\tag{21}
$$

This subproblem mentioned is a projection onto a convex set, ensuring that the pixel values of the restored image remain within the range of 0 to 1.

*3) y-subproblem:* Solving the y-subproblem involves employing a soft thresholding technique coupled with a shrink operator. The formula is presented as follows

$$
\begin{aligned}
y^{k+1}=\arg\max_y&<v^{k+1}\odot o\odot|y|,{\mu_v}^k>+\frac{\beta_v}{2}||v^{k+1}\odot o\odot|y|||^2\\
&+<Hu^{k+1}-b-y,{\mu_y}^k>+\frac{\beta_y}{2}||Hu^{k+1}-b-y||^2,
\end{aligned}
\tag{22}
$$

Eq. (22) holds equivalence to Eq. (23)

$$
\begin{aligned}
y^{k+1}=\arg\max_y&\frac{\beta_y}{2}\left\|y-\left(Hu^{k+1}-b+\frac{\mu_y^k}{\beta_y}\right)\right\|^2\\
&+\frac{\beta_v}{2}\left\|v^{k+1}\odot o\odot|y|+\frac{\mu_v^k}{\beta_v}\right\|^2,
\end{aligned}
\tag{23}
$$

By expanding(23) and discarding the constant terms, we can rewrite (23) as

$$
\begin{aligned}
y^{k+1}=\arg\min_y&\frac{1}{2}\left\|y-\frac{\beta_y(Hu^{k+1}-b+\frac{\mu_y^k}{\beta_y})}{\beta_y+\beta_v(v^{k+1}\odot o)^2}\right\|^2\\
&+\frac{v^{k+1}\odot o\odot\mu_v^k}{\beta_y+\beta_v(v^{k+1}\odot o)^2}\odot|y|,
\end{aligned}
\tag{24}
$$

The minimizer is determined through the application of a four-dimensional shrinkage operator Simplify Eq. (24), we get

$$
y^{k+1}=\text{shrink}\left(\frac{\beta_y(Hu^{k+1}-b+\frac{\mu_y^k}{\beta_y})}{\beta_y+\beta_v(v^{k+1}\odot o)^2},\frac{v^{k+1}\odot o\odot\mu_v^k}{\beta_y+\beta_v(v^{k+1}\odot o)^2}\right),
\tag{25}
$$

where $\text{shrink}(s,\gamma)=\text{sgn}(s)\odot\max\{\|s\|_1-\gamma,0\}$, and $\text{sgn}(\cdot)$ denotes the signum function.

*4) x-subproblem:* Variable $x$ in (26) isupdated by solving the following problem:

$$
x^{k+1}=\arg\min_x\frac{\beta_x}{2}\left\|x-\left(\nabla^\alpha u^{k+1}+\frac{\mu_x}{\beta_x}\right)\right\|^2+\lambda_1||x||.
\tag{26}
$$

With the aid of the previously mentioned shrinkage operator, the $x$-subproblem can be directly tackled.

*5) z-subproblem:* The $z$-subproblem is a recursive filtering issue, with the specific form as follows

$$
z^{k+1}=\arg\min_z\frac{\beta_z}{2}\|z-(u+\mu)\|+\lambda_2\phi(z),
\tag{27}
$$

we define $\vartheta=\sqrt{\frac{\lambda_2}{\beta_z}}$ and $\widetilde{z}^{(k)}=u+\mu$, substitute it into Eq. (27).

$$
z^{k+1}=\arg\min_z\frac{1}{2\vartheta^2}\left\|z-\widetilde{z}^{(k)}\right\|+\phi(z),
\tag{28}
$$

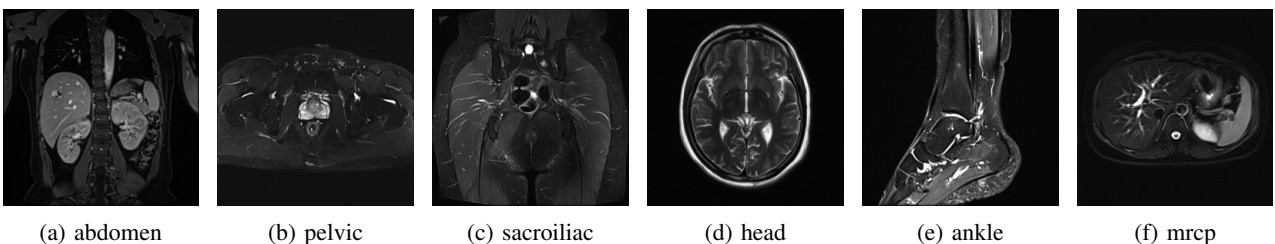

| (a) abdomen | (b) pelvic | (c) sacroiliac | (d) head | (e) ankle | (f) mrcp |

Fig. 1: MR images used for the experiments.

where $\widetilde{z}^{(k)}$ donates the 'blurry' image, (28) minimizes the residue between $\widetilde{z}^{(k)}$ and the 'clean' image $z$ by employing the prior $\phi(z)$.

Expanding on this insight, [19] proposed that the ADMM algorithm can be executed without the prior specifification of $\phi$. Consequently, we may utilize recursive filtering for the resolution of (28). denoted by $\phi$

$$z^{k+1} = \phi_{\vartheta}\left(\widetilde{z}^{(k)}\right). \tag{29}$$

Finally, the Lagrange multipliers are updated by the following

$$\begin{cases} \mu_x^{k+1} = \mu_x^k + \beta_x(\nabla^{\alpha} u^{k+1} - x^{k+1}) \\ \mu_y^{k+1} = \mu_y^k + \beta_y(Hu^{k+1} - b - y^{k+1}) \\ \mu^{k+1} = \mu_z^k + \left(u^{k+1} - z^{k+1}\right) \\ \mu_v^{k+1} = \mu_v^k + \beta_v\left(v^{k+1} \odot o \odot |y^{k+1}|\right) \end{cases} \tag{30}$$

---

**Algorithm 2** solving the minimization problem (13).

**Input:** Regularization parameter $\lambda_2 > 0$, $\alpha$, penalty parameters $\beta_x, \beta_y, \beta_z, \beta_v > 0$, number of iterations. **Initialize:** Initial image $u^0 = b$, counter $k = 0$, Lagrange multipliers $\mu_x, \mu_y, \mu, \mu_v$.
**Output:** Restored image $u$.

1: **for** $k = 0$ **to** number of iterations **do**
2:   // Solve the subproblems:
3:   Compute $u^{k+1}$ according to Eq. (18)
4:   Compute $v^{k+1}$ according to Eq. (19)
5:   Compute $y^{k+1}$ according to Eq. (22)
6:   Compute $x^{k+1}$ according to Eq. (26)
7:   Compute $z^{k+1}$ according to Eq. (28)
8:   // Update the Lagrange multipliers:
9:   Lagrange multipliers according to Eq. (30)
10:   // Check stopping criterion:
11:   **if** $\frac{\|u^{k+1}-u^k\|_2}{\|u^k\|_2} \leq 1 \times 10^{-4}$ **then**
12:     **break**
13:   **end if**
14: **end for**

---

Our method is systematically presented in Algorithm 2. Additionally, there are two remarks that are worth noting regarding this algorithm.

*Remark 1:* When $\alpha = 1$, The fractional-order TV degrades to the conventional TV, then the Eq. (26) reduces to the conventional total variation regularization issue. In the experimental section, we provide an in-depth analysis of how the value of $\alpha$ impacts the noise reduction capabilities of our proposed model.

*Remark 2:* When $\phi(u) = \|u\|_{TV}$ (the total variation norm), then the Eq. (28) is a canonical total variation-based image denoising task [20].

## IV. NUMERICAL EXPERIMENTS

In this section, we exhibit experimental results that verify the effectiveness of our proposed method for image restoration. The real MR test images are shown in Fig 1. The experiment is under Windows 10 and MATLAB R2019b operating system, and the CPU is i5-8250U. The restored image quality was assessed through the peak signal-to-noise ratio (PSNR) and the structural similarity index (SSIM) [19]. Higher PSNR and SSIM values indicate better image quality.

Our experiments employed a relative error-based stopping criterion for the algorithm.

$$RelError = \frac{\|u_{k+1} - u_k\|}{\|u_k\|} \leq 1 \times 10^{-4} \tag{31}$$

where $u_{k+1}$ and $u_k$ are the restored image at the current iterate and previous iterate respectively.

### A. Parameter Selection

The performance of the model is influenced by a number of primary parameters, including the fractional order $\alpha$, the parameters $\lambda_1$, $\lambda_2$, and penalty parameters of $\beta_v, \beta_x, \beta_y$, and $\beta_z$, additionally, these parameters must be meticulously adjusted to achieve higher precision in the outcomes.

Here, we primarily discuss the FOTV term parameter $\alpha$ and the RF parameter $\lambda_2$. Firstly, the value selected for parameter $\alpha$ is crucial importance and the fractional order $\alpha$ is $1 \leq \alpha < 2$. In Fig.2 the 'head' image was processed with a $5 \times 5$ Gaussian blur kernel ($\sigma$=5, impulse=70%) while the 'abdomen' image was subjected to a $7 \times 7$ Gaussian blur kernel ($\sigma$=10, impulse=50%). The figures show the PSNR and SSIM values increase with the $\alpha$ value. Therefore, for Gaussian blur, a value of $\alpha = 1.9$ can obtain the best PSNR and SSIM results.

In Fig.3, the 'ankle' image was processed with a $5 \times 5$ average blur kernel ( impulse=70%) while the 'head' image was subjected to a $7 \times 7$ average blur kernel ( impulse=50%). Our analysis of the images show that the optimal values for PSNR and SSIM are achieved at $\alpha = 1.3$ value, thus our preference for this value in the context of average blur kernel.

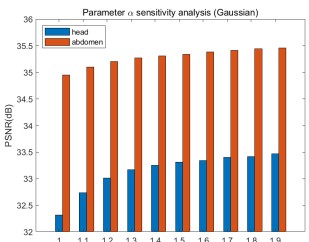
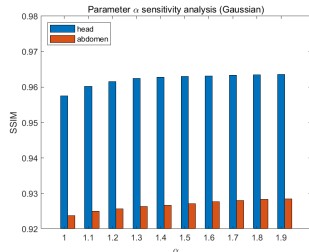

Fig. 2: PSNR and SSIM values for images restoration by my method with different $\alpha$, 'head' ($5 \times 5$ Gaussian blur kernel, $\sigma$=10 and impulse=70%) and 'abdomen' ($7 \times 7$ Gaussian blur kernel, $\sigma$=10 and impulse=50%).

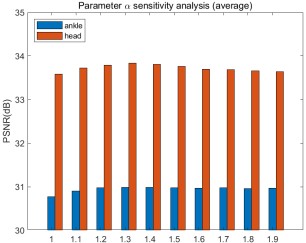
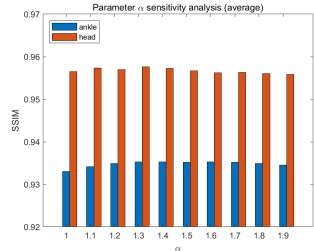

Fig. 3: PSNR and SSIM values for images restoration by my method with different $\alpha$, 'ankle' ($5 \times 5$ average blur kernel and impulse=70%) and 'head' ($7 \times 7$ average blur kernel and impulse=50%).

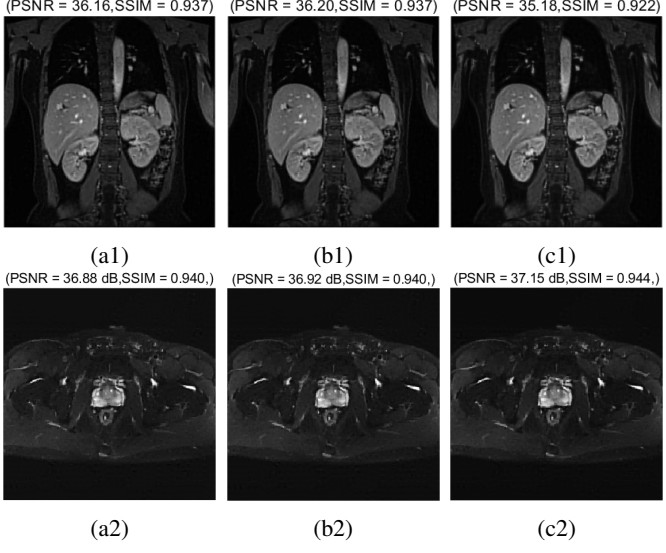

(PSNR = 36.16,SSIM = 0.937)  (PSNR = 36.20,SSIM = 0.937)  (PSNR = 35.18,SSIM = 0.922)

(a1)     (b1)     (c1)

(PSNR = 36.88 dB,SSIM = 0.940,)  (PSNR = 36.92 dB,SSIM = 0.940,)  (PSNR = 37.15 dB,SSIM = 0.944,)

(a2)     (b2)     (c2)

Fig. 4: Restoration results of different $\lambda_2$ for 'abdomen' ($7 \times 7$ Gaussian blur kernel, $\sigma$=10 and impulse=30% ) and 'pelvic' ($7 \times 7$ average blur kernel and impulse=30%).

In this paper, the parameters $\lambda_2$ control the weight and also manage the magnitude of the value $\vartheta$ within the RF. Therefore, the selection of the $\lambda_2$ parameter is crucial. In Fig.4, we conducted experiments with the Gaussian and average blur kernel. The results show that the optimal effect is achieved with $\lambda_2 = 0.001$ when adding a Gaussian blur kernel, and

with $\lambda_2 = 0.005$ when adding an average blur kernel.

In this experimental section, This process involves adjusting one parameter at a time while maintaining the remaining parameters at their default values.

### B. Verify the Effectiveness of the Method

In order to verify the effectiveness of our method in removing staircasing artifacts through fractional-order TV and further noise reduction and texture enhancement through RF, the subsequent sections will validate these capabilities. Our proposed method is called FOTVF (13), and the model that removes the third term $\phi(u)$ from FOTVF is named as FOTV. The model is also solved using the ADMM algorithm.

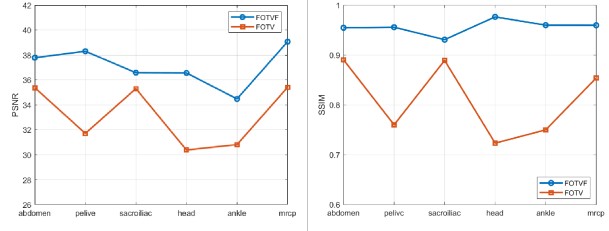

Fig. 5: PSNR and SSIM values of FOTVF item and FOTV restoration results when $5 \times 5$ Gaussian blur kernel (std=5 and impulse=40%).

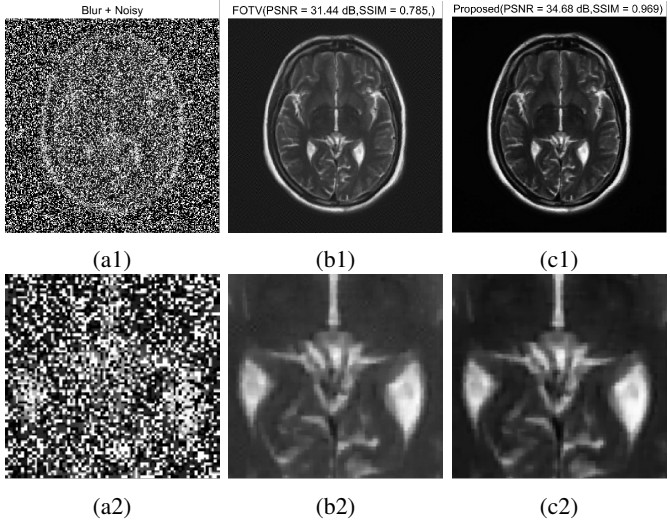

Blur + Noisy FOTV(PSNR = 31.44 dB,SSIM = 0.785,) Proposed(PSNR = 34.68 dB,SSIM = 0.969)

(a1)     (b1)     (c1)

(a2)     (b2)     (c2)

Fig. 6: The first line is the recovered results for 'head' with $5 \times 5$ average blur kernel (std=5 and impulse=60%), while the second line shows the fragments corresponding to the zoomed images. (a1)-(a2) blurry image, (b1)-(b2) FOTV restored, (c1)-(c2) FOTVF restored.

Fig. 5 and Fig. 6 show the PSNR and SSIM values of FOTVF and FOTV at $5 \times 5$ Gaussian blur kernel (std=5 and impulse=40%), and Fig. 6 is the restoration results of FOTVF and FOTV for 'head' at $5 \times 5$ average blur kernel (impulse=60%). It can be seen that the PSNR and SSIM values of FOTV, without the third term decrease a lot. Furthermore, it

TABLE I: Deblurring results for PSNR and SSIM values with $5 \times 5$ Gaussian blur kernel ($\sigma$=5).

| Noise level | Image | Restoration | | | |
| --- | --- | --- | --- | --- | --- |
| | | Median_Filter | HNHOTV-OGS | $\ell_0$-OGSTV | Proposed |
| 30 | abdomen | 28.14/0.756 | 35.69/0.899 | 37.09/0.939 | 38.23/0.958 |
| | pelvic | 27.52/0.855 | 37.62/0.938 | 37.94/0.937 | 38.81/0.957 |
| | sacroiliac | 27.90/0.758 | 35.86/0.912 | 36.23/0.912 | 37.01/0.935 |
| | head | 24.14/0.828 | 34.95/0.904 | 36.85/0.971 | 37.18/0.978 |
| | ankle | 23.53/0.788 | 33.30/0.876 | 35.09/0.956 | 35.15/0.963 |
| | mrcp | 28.62/0.873 | 35.72/0.877 | 37.88/0.932 | 39.47/0.960 |
| 50 | abdomen | 23.70/0.673 | 29.49/0.735 | 36.64/0.938 | 37.26/0.950 |
| | pelvic | 23.61/0.770 | 28.80/0.680 | 37.48/0.941 | 37.55/0.953 |
| | sacroiliac | 23.94/0.680 | 30.90/0.803 | 35.72/0.913 | 36.07/0.925 |
| | head | 21.39/0.744 | 30.25/0.764 | 35.68/0.969 | 35.82/0.974 |
| | ankle | 21.35/0.707 | 28.44/0.670 | 33.61/0.950 | 33.65/0.955 |
| | mrcp | 23.82/0.786 | 27.82/0.593 | 37.80/0.939 | 38.58/0.959 |
| 70 | abdomen | 13.42/0.288 | 17.18/0.198 | 35.21/0.930 | 35.27/0.932 |
| | pelvic | 13.51/0.278 | 16.14/0.071 | 35.21/0.940 | 34.89/0.942 |
| | sacroiliac | 13.82/0.245 | 15.07/0.087 | 34.27/0.904 | 34.22/0.902 |
| | head | 13.00/0.333 | 17.00/0.291 | 33.45/0.962 | 33.50/0.964 |
| | ankle | 13.07/0.297 | 16.93/0.220 | 30.69/0.933 | 30.86/0.935 |
| | mrcp | 13.33/0.313 | 17.24/0.105 | 36.50/0.948 | 36.62/0.954 |

TABLE II: Deblurring results for PSNR and SSIM values with $7 \times 7$ Gaussian blur kernel ($\sigma$=10).

| Noise level | Image | Restoration | | | |
| --- | --- | --- | --- | --- | --- |
| | | Median_Filter | HNHOTV-OGS | $\ell_0$-OGSTV | Proposed |
| 30 | abdomen | 26.93/0.686 | 34.98/0.894 | 35.55/0.917 | 36.20/0.937 |
| | pelvic | 26.70/0.825 | 36.40/0.932 | 36.18/0.916 | 36.82/0.943 |
| | sacroiliac | 27.05/0.714 | 34.74/0.897 | 34.67/ 0.887 | 35.12/0.910 |
| | head | 22.50/0.765 | 34.00/0.905 | 34.98/0.962 | 35.00/0.967 |
| | ankle | 22.49/0.732 | 32.15/0.880 | 33.10/0.938 | 32.82/0.944 |
| | mrcp | 27.28/0.835 | 35.49/0.891 | 36.15/0.907 | 37.43/0.944 |
| 50 | abdomen | 23.30/0.616 | 31.99/0.800 | 34.92/0.910 | 35.46/0.928 |
| | pelvic | 23.32/0.745 | 31.11/0.791 | 35.73/0.921 | 36.08/0.940 |
| | sacroiliac | 23.63/0.642 | 33.16/0.865 | 34.18/0.880 | 34.36/0.899 |
| | head | 20.59/0.691 | 31.46/0.802 | 34.11/0.957 | 33.89/0.961 |
| | ankle | 20.79/0.660 | 29.43/0.740 | 31.91/0.931 | 31.62/0.935 |
| | mrcp | 23.41/0.755 | 30.77/0.727 | 35.90/0.914 | 36.78/0.943 |
| 70 | abdomen | 13.40/0.266 | 18.67/0.228 | 34.07/0.904 | 34.08/0.909 |
| | pelvic | 13.50/0.268 | 17.20/0.086 | 34.56/0.924 | 34.46/0.929 |
| | sacroiliac | 13.80/0.230 | 16.60/0.139 | 33.30/0.876 | 33.34/0.881 |
| | head | 12.93/0.308 | 19.47/0.355 | 32.49/0.948 | 32.01/0.945 |
| | ankle | 13.02/0.275 | 20.02/0.291 | 30.08/0.917 | 29.77/0.915 |
| | mrcp | 13.31/0.300 | 19.28/0.141 | 35.17/0.926 | 35.43/0.941 |

TABLE III: Deblurring results for PSNR and SSIM values with $5 \times 5$ average blur kernel.

| Noise level | Image | Restoration | | | |
|---|---|---|---|---|---|
| | | Median_Filter | HNHOTV-OGS | $\ell_0$-OGSTV | Proposed |
| 30 | abdomen | 28.30/0.753 | 35.54/0.897 | 36.92/0.937 | 37.31/0.950 |
| | pelvic | 28.13/0.854 | 37.70/0.940 | 37.90/0.936 | 38.80/0.956 |
| | sacroiliac | 28.18/0.756 | 35.95/0.913 | 36.26/0.913 | 37.07/0.933 |
| | head | 24.08/0.825 | 34.92/0.899 | 36.89/0.971 | 37.02/0.977 |
| | ankle | 23.70/0.786 | 33.33/0.873 | 35.27/0.955 | 35.32/0.964 |
| | mrcp | 28.70/0.871 | 35.64/0.876 | 37.81/0.927 | 39.63/0.964 |
| 50 | abdomen | 23.73/0.671 | 29.25/0.733 | 36.70/0.939 | 36.40/0.942 |
| | pelvic | 23.76/0.770 | 27.93/0.651 | 37.35/0.941 | 37.65/0.952 |
| | sacroiliac | 24.00/0.678 | 29.79/0.776 | 35.94/0.917 | 36.17/0.925 |
| | head | 21.35/0.742 | 29.89/0.751 | 35.82/0.969 | 35.69/0.973 |
| | ankle | 21.42/0.705 | 28.55/0.673 | 33.56/0.950 | 33.68/0.955 |
| | mrcp | 23.81/0.785 | 27.29/0.571 | 37.82/0.939 | 38.66/0.961 |
| 70 | abdomen | 13.42/0.287 | 16.94/0.189 | 35.43/0.932 | 34.49/0.920 |
| | pelvic | 13.52/0.277 | 15.65/0.061 | 35.06/0.941 | 35.34/0.943 |
| | sacroiliac | 13.82/0.245 | 14.83/0.075 | 34.48/0.906 | 34.23/0.902 |
| | head | 13.00/0.332 | 17.19/0.295 | 33.58/0.963 | 33.25/0.962 |
| | ankle | 13.08/0.296 | 16.81/0.209 | 30.87/0.935 | 30.99/0.935 |
| | mrcp | 13.33/0.313 | 17.38/0.101 | 36.61/0.948 | 36.74/0.956 |

TABLE IV: Deblurring results for PSNR and SSIM values with $7 \times 7$ average blur kernel.

| Noise level | Image | Restoration | | | |
|---|---|---|---|---|---|
| | | Median_Filter | HNHOTV-OGS | $\ell_0$-OGSTV | Proposed |
| 30 | abdomen | 26.89/0.684 | 34.92/0.892 | 35.32/0.912 | 35.53/0.927 |
| | pelvic | 26.68/0.824 | 36.45/ 0.932 | 36.08/0.913 | 37.15/0.944 |
| | sacroiliac | 27.03/0.712 | 34.81/ 0.898 | 34.21/0.866 | 35.42/0.910 |
| | head | 22.45/0.763 | 34.09/0.906 | 35.32/0.961 | 35.00/0.964 |
| | ankle | 22.46/0.730 | 32.26/0.880 | 33.36/0.937 | 33.38/0.946 |
| | mrcp | 27.24/0.833 | 35.48/0.892 | 35.65/0.890 | 37.77/0.951 |
| 50 | abdomen | 23.29/0.614 | 32.03/0.801 | 35.03/0.912 | 34.76/0.917 |
| | pelvic | 23.31/0.744 | 31.08/0.788 | 35.93/0.922 | 36.36/0.941 |
| | sacroiliac | 23.62/0.641 | 33.13/0.865 | 34.24/0.882 | 34.68/0.900 |
| | head | 20.57/0.689 | 31.40/0.795 | 34.24/0.956 | 33.84/0.958 |
| | ankle | 20.77/0.658 | 29.47/0.737 | 32.17/0.931 | 32.17/0.937 |
| | mrcp | 23.40/0.754 | 30.57/0.721 | 35.89/0.909 | 37.04/0.948 |
| 70 | abdomen | 13.39/0.266 | 18.76/0.229 | 34.21/0.908 | 33.38/0.921 |
| | pelvic | 13.50/0.267 | 17.09/0.084 | 34.78/0.925 | 34.85/0.943 |
| | sacroiliac | 13.80/0.230 | 16.45/0.137 | 33.43/0.877 | 34.48/0.883 |
| | head | 12.93/0.307 | 19.64/0.356 | 32.58/0.947 | 32.03/0.945 |
| | ankle | 13.02/0.274 | 19.80/0.290 | 30.23/0.918 | 30.08/0.917 |
| | mrcp | 13.31/0.300 | 19.16/0.135 | 35.40/0.927 | 35.67/0.944 |

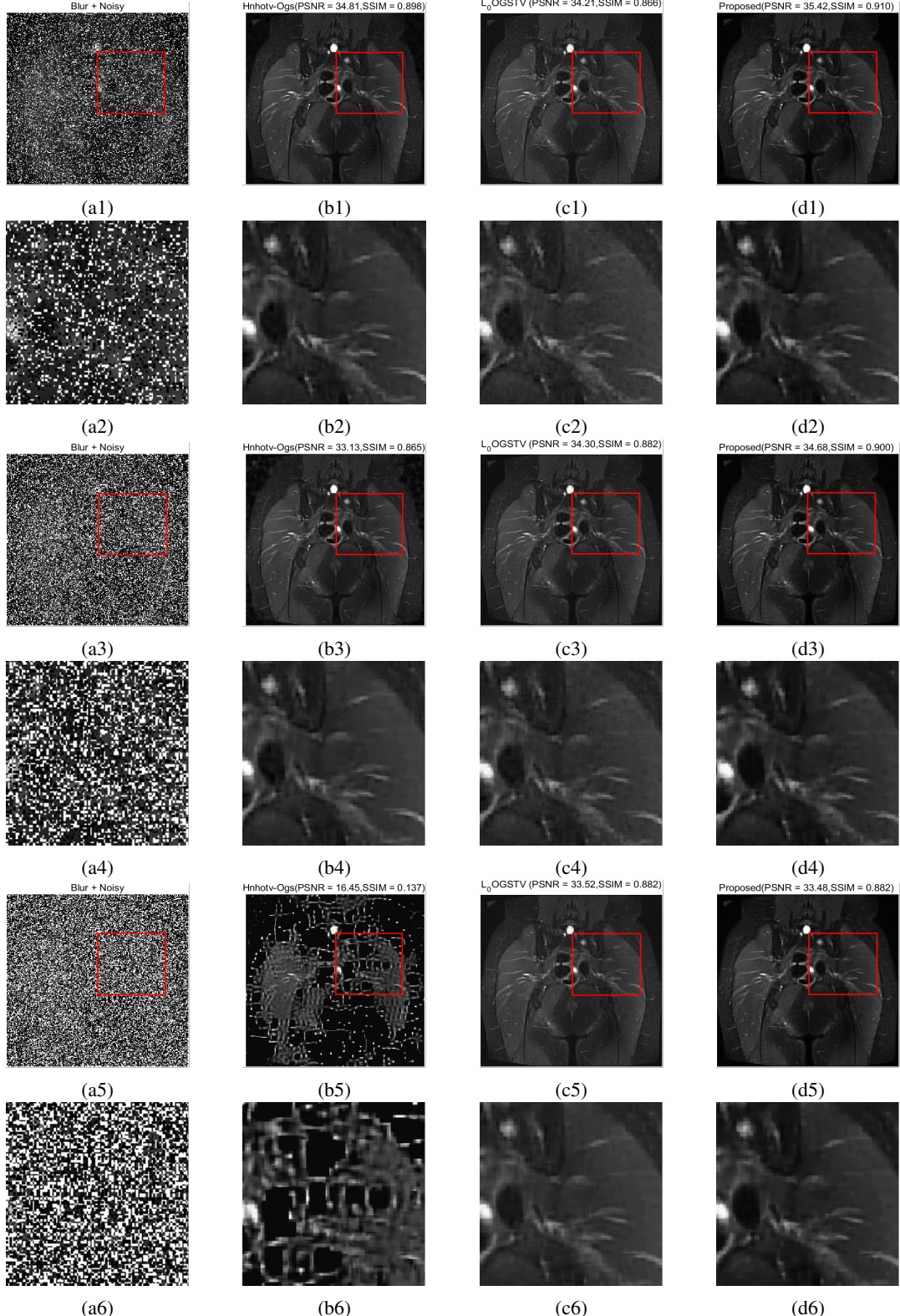

Fig. 7: The (1), (3), and (5) rows show the results after applying a $7 \times 7$ average blur kernel ( impulse=$30\%, 50\%, 70\%$) to the 'sacroiliac' image. The (2), (4), and (6) rows display the corresponding enlarged image segments. Columns (a1)-(a6) show the noisy versions, (b1)-(b6) are restored with HNHOTV-OGS, (c1)-(c6) with $\ell_0$-OGSTV, and (d1)-(d6) with our method.

can be observed from these figures that the restoration effect of the FOTV is not ideal. In contrast, the FOTVF recovery image not only smooths out the staircase artifacts but also achieves a higher SSIM value.

### C. Results and Analysis

The experimental results of our proposed model are compared with three related methods: $\ell_0$-OGSTV [14], HNHOTV-OGS [3], and Median Filter (built-in noise reduction technology in MATLAB).

In this experiment, we assumed that the blurring kernel is known. To simulate a noisy blurred image, we blur the original images with a $5 \times 5$ Gaussian blur kernel with standard deviation $\sigma = 5$ and a $7 \times 7$ Gaussian blur kernel with standard deviation $\sigma = 10$, or two different sizes of average blur kernel $5 \times 5$ and $7 \times 7$ to the original images. Then, we introduce impulse noise with different densities to the blurred images.

For parameters, we fixed $\lambda_1 = 1$. The other parameters were manually selected to obtain the most satisfactory restoration quality. The parameters of HNHOTV-OGS and $\ell_0$-OGSTV are consistent with the original text. Three different impulse noise levels of 30%, 50% and 70% are added to the test image respectively to generate each observed image. The obtained PSNR and SSIM values are shown in Table I to Table IV.

In each table, we can observe when using various intensities of noise and different blur kernels, our method almost always achieves higher PSNR and SSIM values compared to the other methods. Only in a few specific cases does $\ell_1$-OGSTV perform slightly better. From Table.I to Table.IV, it can be observed that our method performs well when relatively low levels of noise, Despite a modest reduction in denoising efficacy compared to $\ell_0$-OGSTV under conditions of high-density noise, it continues to deliver superior PSNR and SSIM values.

For the Fig.7, we present the outcomes of three distinct denoising models applied to MR images. The image 'sacroiliac' is treated with average blur and 30%, 50%, and 70% impulse noise. In the results, the HNHOTV-OGS image denoising effect is not satisfactory; there are slight blocky artifacts, and the denoising performance is poor at higher noise densities.

The main competition for our technique is the $\ell_0$-OGSTV model, which effectively eliminates impulse noise and mitigates staircase effects through the application of the $\ell_0$-norm. Despite the addition of varying degrees of Gaussian kernels and average kernels, our method consistently outperformed others, maintaining its advantage in producing better results.

### V. CONCLUSION

We proposed a MR image restoration model that combines regularization and filtering methods, aiming at effectively removing impulse noise and staircasing artifacts present in MR images. We demonstrated the effectiveness of using the $\ell_0$-norm as a data fidelity term to eliminate impulse noise, while incorporating fractional-order total variation and recursive filtering as penalty terms to mitigate staircasing artifacts and preserve important edges. We solved the proposed model using the alternating direction method of multipliers.

In experiments involving Gaussian and average blurring, our method outperformed three other methods in terms of PSNR and SSIM across various levels of blur and noise.

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
