# OpenReview forum: "MR Image Restoration by Utilizing Fractional-Order TV and Recursive Filtering"
_IEEE.org/ICIST/2024/Conference — IEEE ICIST 2024 Conference Submission_

### Official Review · Reviewer_npZ8 · 2024-08-22
**Well-founded approach**

**Rating:** 7
**Confidence:** 3

**Review:**

The paper presents a novel and well-founded approach to MR image restoration, combining fractional-order TV regularization with recursive filtering. This topic is interesting, the following comments need to further consider: (1) Simplify the language in the abstract and introduction, and provide brief explanations for technical terms such as "fractional-order TV" and "recursive filtering." This will help readers from different backgrounds understand the significance of the research. (2) Provide a brief summary of how the proposed method compares to existing approaches in terms of PSNR and SSIM. Mentioning the specific methods used for comparison and highlighting key results would provide context and reinforce the paper's contributions. (3) Include a brief discussion on the assumptions made in the model, such as the conditions under which the method is expected to perform best. Additionally, acknowledging any limitations of the approach would present a more balanced view of the research. (4) Discuss the potential practical applications of the method in more detail.

---

### Official Review · Reviewer_zGY3 · 2024-08-22
**Manuscript Accept**

**Rating:** 7
**Confidence:** 3

**Review:**

Does the proposed MR image restoration model address the effect of the noise levels with high density?
What is the specific formula of the stopping criterion?  Algorithms 1 and 2 should keep a uniform format.
What is the difficulty when the authors make the combination of ℓ0-norm data fidelity, fractional-order total variation and recursive filtering?

---

### Official Review · Reviewer_crzw · 2024-08-23
**This is a good paper.**

**Rating:** 8
**Confidence:** 4

**Review:**

This paper proposes a novel MR image restoration model incorporating fractional-order regularization and filtering methods, which eliminate impulse noise and preserves structural information while mitigating staircase artifacts during deblurring. Finally, this paper solves the corresponding optimization model by using the alternating direction method of multipliers. Experimental results demonstrate the effectiveness of the proposed method in restoring medical images.
1). Please compare with existing papers to highlight the innovative of this article.
2). Since the value chosen for parameter $alpha$  is crucial, provide a guide to finding the optimal parameter $alpha$.

---

### Decision · Program_Chairs · 2024-09-08

Accept (Oral)